# The Dandarah App: An mHealth Platform to Tackle Violence and Discrimination of Sexual and Gender Minority Persons Living in Brazil

**DOI:** 10.3390/ijerph20010280

**Published:** 2022-12-24

**Authors:** Angélica Baptista Silva, Mônica Malta, Cosme Marcelo Furtado Passos da Silva, Clarice Cavalcante Kalume, Ianê Germano Andrade Filha, Sara LeGrand, Kathryn Whetten

**Affiliations:** 1Department of Human Rights, Health and Cultural Diversity, National School of Public Health, Oswaldo Cruz Foundation, Rio de Janeiro 21040-360, Brazil; 2Institute for Mental Health Policy Research, Centre for Addiction and Mental Health, Toronto, ON M5S 2S1, Canada; 3Department of Psychiatry, Faculty of Medicine, University of Toronto, Toronto, ON M5T 1R8, Canada; 4Department of Epidemiology and Quantitative Methods in Health, National School of Public Health, Oswaldo Cruz Foundation, Rio de Janeiro 21040-360, Brazil; 5Center for Health Policy and Inequalities Research, Duke Global Health Institute, Duke University, Durham, NC 27710, USA

**Keywords:** sexual and gender minority (SGM), LGBTQ, LGBT, violence, sexual violence, domestic violence, discrimination, gender-based violence, violence, mHealth, app, digital health, mobile health

## Abstract

Discrimination and violence are widely experienced by sexual and gender minority (SGM) persons worldwide. More than one SGM person is murdered every day in Brazil because of their sexuality or gender identity, which is the highest reported homicide rate in the world. Alt-hough discrimination and violence against SGM persons in Brazil are considered to be hate crimes, reporting is still suboptimal due to fear of police SGM phobia and victim blaming. Accessible and easily disseminated interventions are urgently needed. Herein, we describe the develop-ment of an mHealth solution to help address violence against SGM persons, namely the Rainbow Resistance: Dandarah App, with a synthesis of key results and feedback from the SGM community after 24 months of using the app. Twenty-two focus group discussions (FGDs) were conducted with SGM persons living in six Brazilian states: Bahia, Federal District, São Paulo, Rio de Janeiro, Minas Gerais, Sergipe, and Pará. A total of 300 SGM persons participated in the FGDs. A thematic analysis was performed to interpret the qualitative data. Content themes related to aesthetics, us-ability, barriers to resources, and likes/dislikes about the intervention arose from the FGDs. Participants found the intervention to be user-friendly, endorsed more likes than dislikes, and suggested a few changes to the app. The findings suggest that the intervention is usable and fit for future ef-fectiveness testing, and that it could fill an important gap in the well-being of SGM persons living in a country with high levels of discrimination and violence towards this community, i.e., Brazil.

## 1. Background

According to the WHO, violence is the “the intentional use of physical force or power, threatened or actual, against oneself, another person, or against a group or community, that either results in or has a high likelihood of resulting in injury, death, psychological harm, maldevelopment, or deprivation” [1]. In addition, the principle of non-discrimination seeks “… to guarantee that human rights are exercised without discrimination of any kind based on race, color, sex, language, religion, political, or other opinion, national or social origin, property, birth or other status such as disability, age, marital and family status, sexual orientation and gender identity, health status, place of residence, economic and social situation” [2].

Sexual and gender minority (SGM) persons, i.e., lesbian, gay, bisexual, transgender, non-binary, queer, questioning, and intersex persons, are a highly diverse and multifaceted population. Although the SGM community has become increasingly visible, engaged, and mobilized in Brazil, there have been uneven gains in advancing their well-being and health status, including reducing experiences of violence and discrimination, and a range of health disparities remains prevalent [3,4,5]. Violence and discrimination against SGM persons is a key human rights violation, has been reported in all countries, seems to have increased during the COVID-19 pandemic, and is a core reason for the health disparities identified among this population [6,7,8,9,10].

In 2019, Brazil’s Supreme Court voted to criminalize homo- and transphobia, deciding that discrimination based on sexual orientation or gender identity violates the nation’s constitution [11,12]. However, daily violence against SGM persons continues to be highly prevalent in the country. More than one SGM person is murdered every day in Brazil because of their sexuality and/or gender identity, which is the highest reported homicide rate in the world [13]. According to worldwide data collected by Transgender Europe (TGEU) since 2008, Brazil is responsible for 41.5% of all reported murders among trans and gender-diverse people in the world [14].

However, the above-mentioned unacceptable numbers are just the tip of the iceberg. Frequently, violent crimes and other hate incidents against SGM persons are not reported or prosecuted because of chronic distrust between the SGM community and police. For example, the national health system in Brazil lacks information about gender identity and sexual orientation, making the majority of SGM-related crimes invisible. This scenario precludes the development of population-based studies that could ground more inclusive and responsive policies.

Daily experiences of stigma, prejudice, and discrimination create a hostile and stressful social environment that causes mental health problems among SGM persons worldwide. In Brazil, where a scenario of widespread violence and impunity is present, a high prevalence of anxiety, depression, suicidality, and PTSD have also been reported among SGM persons [15,16,17].

Several mobile health (mHealth) solutions have been developed to address gender-based violence. As defined by the World Health Organization (WHO), mHealth is an area of electronic health that provides health services and information to both patient populations and healthcare providers via mobile technologies such as mobile phones, especially smartphones [18]. Mobile health can offer tools to enable data exchange and storage, remote data capture, and sharing of relevant information across the health ecosystem. According to the Global Strategy on Digital Health [19], it is part of a disruptive digital transformation of healthcare with technologies such as the Internet of things (IoT refers to the billions of physical devices around the world that are now connected to the internet), virtual care, remote monitoring, artificial intelligence, big data analytics, blockchain, and intelligent wearables, thus, creating more evidence-based knowledge, skills, and competence for professionals to support health care,.

Herein, we describe the development of an mHealth solution designed to address some aspects of violence against SGM persons, namely the Rainbow Resistance: Dandarah App, with a synthesis of key results and feedback from the the SGM community after 24 months of using the app.

## 2. Methods

### 2.1. Approach

We implemented a community-based participatory research (CBPR) methodology to develop and to pilot and evaluate the Rainbow Resistance: Dandarah App. CBPR is a research strategy that includes collective, reflective, and systematic actions, in which researchers and community stakeholders work as equal partners in all steps of the research process [20,21]. CBPR’s major goal is to inform interventions and practices that bring social change to a specific community group; in our study, the main goal was to tackle some aspects of violence against the SGM community in Brazil.

Our study included two key phases: Phase 1 focused on the development of the Rainbow Resistance: Dandarah App, which included working with the SGM community to identify their priorities and which features should be included in this mHealth app. Phase 2 focused on piloting the Rainbow Resistance: Dandarah App with members of the Brazilian SGM community. The project was approved by the Sergio Arouca National School of Public Health (CAAE:22881019.4.0000.5240) from the Oswaldo Cruz Foundation, and all participants in the study provided written informed consent.

#### 2.1.1. Phase 1: Development of the “Rainbow Resistance: Dandarah App”

Congressman Jean Wyllys, one of Brazil’s first openly gay congressmen, allocated federal funding towards a project tackling violence against the SGM community. Our research team worked together with the SGM community leaders to identify their priorities, and to develop the strategy and which features could allow the SGM community to map and record episodes of discrimination and violence. This mHealth strategy included the development of a smartphone app entitled the Rainbow Resistance: Dandarah App, in memory of Dandara dos Santos, a transgender woman brutally murdered in 2017 [22].

We started conducting in-depth interviews with SGM key informants to map the community needs and to identify features that should be included in the Rainbow Resistance: Dandarah App. Then, a multidisciplinary team that included public health and health informatics experts, software/website developers, and representatives of the SGM community collaborated to develop the Rainbow Resistance: Dandarah App.

Two apps were developed, one app for iPhone users and another app for Android users; both apps addressed the same goals, preferences, and needs identified by SGM key informants. The final pilot app included a mapping tool that identified safe and unsafe areas, as reported by users and a panic button that, if pushed, would send a text message to persons previously registered by each app user. This panic button identified the geolocation of where it was pushed so that those previously identified as community members and allies who are willing to help the person could get to the place where the violence was happening in real time. Victims of discrimination and/or violence received prompt follow-up and referrals when needed. 

Additional strategies included the development of a study website (resistenciaarcoiris.ensp.fiocruz.br) and the utilization of Twitter, Facebook, YouTube, and Instagram to inform the Brazilian SGM community about legislation related to discrimination and violence. This broad strategy allowed the identification of a large network of supportive services available in each Brazilian state for the SGM community. SGM representatives were also trained to develop and format websites and to produce and format videos.

#### 2.1.2. Phase 2: Pilot Test, Refining, and Improvement of the Rainbow Resistance: Dandarah App

##### Focus Group Discussions

Local leaders and SGM organizations were approached in six Brazilian states: Bahia, Federal District, São Paulo, Rio de Janeiro, Minas Gerais, Sergipe, and Pará. The selected states represent a mix of large, urban areas (São Paulo and Rio de Janeiro), mixed urban and rural (Minas Gerais and Bahia), mostly rural areas (Sergipe and Pará), and the Federal District. Data on reported crimes show a higher prevalence of violence in rural, hard-to-reach states such as Pará and Sergipe [23]. Field team supervisors from each state selected representatives from the local SGM community, with the aim of recruiting focus group participants who reflected the diversity of culture and conditions in each community, taking into consideration race, ethnicity, gender, age, and education. 

Possible participants of the focus group discussions (FGDs) were approached first by phone, followed by email. The initial phone call was important to thoroughly explain the study objectives and to discuss any concerns. For those interested in participating, an email would follow with the study description and link to download the Rainbow Resistance: Dandarah App.

The FGDs were facilitated by a moderator, and a notetaker who recorded hand-written notes and observations of nonverbal communication and other interactions among participants. Each FGD had between 12 and 15 participants, and each state had at least one FGD comprised exclusively of transgender and gender non-conforming persons, and other FGDs with SGMs in general. This strategy aimed at addressing the specific needs of transgender and gender non-conforming persons, which is a group at higher risk for reported discrimination and violence in Brazil as compared with other SGM persons. At the beginning of each FGD, participants were requested to download the Rainbow Resistance: Dandarah App and test its features.

The FGDs were semistructured, lasted ~90 min, and used a script that included probes about: (1) experiences of discrimination and violence among participants; (2) willingness and perceived efficacy of an mHealth strategy to tackle discrimination/violence against SGMs; (3) suggestions of features that should be included on the App; (4) perceptions about the App, including critiques and proposals to improve it; (5) challenges to use the app to map and report discrimination/violence, and strategies to overcome those difficulties.

The final version of the Rainbow Resistance: Dandarah App included FGD findings, and several features were revised/improved following the FGD results. The intervention was launched in December 2019, and in 2020, it became a strategy sponsored and maintained by the Brazilian Ministry of Health. This allowed for a close monitoring of the App, prompt resolution of bugs and security flaws, as well as great responsiveness and communication with users. The characteristics of the Rainbow Resistance: Dandarah App users are presented in absolute/relative frequencies.

This intervention strategy was registered at the Brazilian National Institute of Intellectual Property, and three patents were issued for this intervention: Rainbow Resistance: Dandarah App for Android, Rainbow Resistance: Dandarah App for IOS, and Rainbow Resistance: Dandarah website. This initiative became a governmental intervention, sponsored by the Sergio Arouca School of Public Health, from the Brazilian Ministry of Health.

#### 2.1.3. Theoretical Framework

The development of the Rainbow Resistance: Dandarah App was guided by social cognitive theory (SCT), which examined the process by which personal factors, environmental factors, and human behavior influence each other [24,25]. Following SCT, the app included features targeting goal setting, self-regulatory functions (e.g., self- monitoring selected safe/risky places for SGM persons), and self-efficacy (e.g., ability to avoid dangerous places (informed by crowdsourcing)). These features were enhanced by environmental conditions, such as lists of legislation protecting SGM persons and targeting hate crimes as well as addresses of shelters, NGOs, and healthcare services.

Informed by SCT, goal setting was targeted by encouraging users to identify safe spaces (e.g., SGM-appropriate services, NGOs, shelters) and violent incidents through a system-prompted (cued data entry) and non-prompted (user-prompted data entry) approach to generate a more comprehensive map, including a broad range of reports. Goal setting was targeted through the development of a personalized safety plan with routes to avoid (based on recent violence reports) and SGM-friendly services nearby, as informed by crowdsourcing.

Self-regulation and self-efficacy were addressed with features developed to enhance the feeling of control over one’s ability to avoid previously reported dangerous places and to create a personalized plan if the place could not be avoided (e.g., share with trusted peers their location, ask for a ride if needed). A critical feature was the ability to use the app to share violent experiences and the panic button that allowed 5 preselected contacts to be immediately informed about the user’s location and dangerous situation.

#### 2.1.4. Analysis

All transcripts were reviewed and independently coded by two investigators (A.B.S. and I.G.A.F.). Themes/coding discrepancies were discussed to reach an agreement between coders. During this process, key domains were continually reassessed and revised. Selected quotes were included to illustrate major research findings reported by FGD participants. The selection of quotes aimed at covering all expressed viewpoints while avoiding redundancy.

We began with a list of content based on existing guidelines and the literature [15,16,17,23] but inductively generated codes from the collected data. Two investigators (A.B.S. and I.G.A.F.) independently reviewed each transcript using the constant comparative method to assign codes [26]. The group met after each transcript to eliminate or refine codes as needed. Disagreements were resolved by negotiated consensus. The investigators continued this process until no new concepts were generated by review of successive transcripts. Then, every transcript was independently recoded by the 2 investigators (A.B.S. and I.G.A.F.) using the final code structure, with disagreements resolved by consensus. The final inter-coder reliability was high (Cohen kappa > 0.80). We used the Atlas software, version 8.4.26.0 (GmbH, Berlin, Germany) to facilitate qualitative analysis. 

## 3. Results

Twenty-two FGDs were conducted with SGM persons living in six Brazilian states: Bahia, Federal District, São Paulo, Rio de Janeiro, Minas Gerais, Sergipe, and Pará. A total of 300 SGM persons participated in the FGDs.

### 3.1. Focus Group Discussion Results

The thematic analysis identified two main themes, mentioned in all FGD: the impact of violence towards SGM persons within the community and the utility of the Rainbow Resistance: Dandarah App to tackle this reality. All participants mentioned that the panic button feature was a key strategy offered by the app, and this was the topic with more intensity, frequency, weight, and importance with 292 citations (Figure 1).

Additional contributions related to SGM violence included the importance to conduct more scientific studies followed by interventions developed to improve the quality of life of SGM persons: *“Hey y’all, queer people are not being studied by academics! How cool is that? Not it’s time for you doctors to actually DO something for us, right?”*

Another key citation refers to the invisibility of the SGM population in healthcare centers and health datasets:


*“… The Health Department keeps telling that we do not use outpatient units at all, there is no demand for additional services for queer people, so there’s no need to improve or increase whatever they have… But actually the Health Department has key network problems, you know? The different systems have A LOT of problems: our data is hidden in isolated databases, systems are incompatible, and the data is difficult to exchange, analyze, interpret… Like, just two weeks ago, our appointments were made on a paper calendar. Did you hear that? On a PAPER calendar. C’mom, it’s not that we do not exist, the Health Department needs to get digital, and then they will see that we’re using their services a lot”.*


The third citation highlights how educated the SGM population is about their rights and Brazilian legislation:


*“And we give them some examples. What is included in the Maria da Penha Law? To punch, slap, and push, among others. Verbal aggression, property violence, sexual violence, institutional violence—that’s all included in the Brazilian Penal Code. We gotta know our rights”.*


Table 1 represents the codes with higher density. Density shows the number of links between codes/themes; a code with high density means that it has a lot of links to other codes. This table was organized to help visualize which aspects need to be revised on the Rainbow Resistance: Dandarah App.

Observations about a possible Internet of Things device that would interact with the system raised important questions (Table 1). The FGD participants also mentioned that a possible ring/bracelet could use near-field communication (NFC). NFC is a short-range wireless connectivity technology that lets NFC-enabled devices to communicate with each other. In our intervention, it could allow a ring/bracelet to communicate with a phone, for example, by pushing the panic button and its connected services. Many participants mentioned a key disadvantage related to earrings and necklaces. Those possible devices could be easily lost during a physical confrontation or lost in bathrooms. Smartwatches could be a better option, especially if a lower cost model that included only a panic button could be developed.

The preferred accessory was a bracelet, and the FGD participants made many design and technology suggestions: luminous; with buttons and a built-in recorder; counting users’ rapid heartbeats and posting threat alerts; with GPS, working independently of the cell phone; of low-cost and connecting with the same database as Dandarah’s panic button. Bracelets were also mentioned as an ideal accessory for SGM persons who were also deaf or hard of hearing people.

### 3.2. Rainbow Resistance: Dandarah App: Revised App

The home page of the Rainbow Resistance: Dandarah App is presented in Figure 2. Following the results from the FGDs, the panic button was moved to the top of the initial screen. App colors, icons, and content were also revised, to include FGD suggestions.

### 3.3. Characteristics of App Users

The characteristics of the Rainbow Resistance: Dandarah App are described in Table 2. Between the app launch on 18 December 2019, and 9 August 2022, 4114 users were registered. The majority of app users are young adults aged 29 or less (60.9%) and white (47.3%). Most app users report their gender identity as cisgender men (50.9%), followed by cisgender women (19.2%). More frequent sexual orientations among app users include gay (43.5%), followed by bisexual (14.6%) and lesbian (12.8%). A small percentage of participants auto-identify as intersex (6.8%), and less than 3% mention being a person living with a disability (Table 2).

## 4. Discussion

The current study presented the development of the Rainbow Resistance: Dandarah App, an mHealth application for SGM from Brazil. It was developed to address the high prevalence of discrimination and violence experienced by the Brazilian SGM community [23,27].

Similar to previous research, study participants also indicated that they had experienced additional victimization and SGM phobia when they reported harassment/abuse to the police [28,29]. Unfortunately, SGM awareness training is not offered or required by Brazilian police officers. SGM persons also experience significant barriers to accessing health, psychological and social services, as well as SGM-friendly shelters after sexual/physical assaults [30].

Therefore, the Rainbow Resistance: Dandarah App may be a feasible strategy to address barriers to reporting assault and accessing follow-up care among SGM persons from Brazil. The mHealth initiative can be freely accessed at any time, is supported by the Brazilian Ministry of Health, and connects users with services nearby their location right after they press the panic button. More importantly, through crowdsourcing and mapping, the Rainbow Resistance: Dandarah App allows users to highlight services that are more welcoming for SGM persons, therefore, generating a list of services that the SGM community has already used and approved. This additional strategy is key to improving assault survivors’ willingness to look for treatment and care after rape, physical assault, and other forms of violence, knowing that their sexual orientation and gender identity will be valued and respected.

Overwhelmingly, the participants indicated that the panic button feature is a helpful resource for SGM persons who experience assaults due to its unique options to connect app users in real time with a supportive network of friends and trusted allies. Participants who tested the Rainbow Resistance: Dandarah App provided useful feedback regarding the usability of the application, as well as key suggestions that were incorporated (or will be included soon).

Regarding the app’s aesthetics and usability, participants indicated that the Rainbow Resistance: Dandarah App was easy to use, its features were clear, the available information was not overwhelming, and the layout was appropriate and interesting to catch the attention of SGM persons. Suggestions for improvement included moving the panic button up on the screen and providing mental health interventions for those surviving assaults. The first suggestion was implemented and the second suggestion is being developed within a follow-up study. Overall, the study findings suggested that the Rainbow Resistance: Dandarah App usability was successful. It will be interesting to evaluate the app’s efficacy in another study.

Although participants indicated that the Rainbow Resistance: Dandarah App was discreet and the additional bracelet was perceived as a helpful tool during an assault, additional precautions may be needed to protect the confidentiality of the information stored on the Rainbow Resistance: Dandarah App. For SGM persons who are sex workers, experiences of physical and sexual violence are highly frequent, and access to healthcare is suboptimal [31]. Moreover, SGM persons who are sex workers might be financially dependent on their pimps and clients. Therefore, any breach of privacy while using the Rainbow Resistance: Dandarah App could present a life-threatening episode [31]. In the specific context of intimate partner violence (IPV), some studies have found a greater prevalence of IPV among SGM partners as compared with heterosexual couples, while their likelihood of engaging in follow-up services is suboptimal [32]. It is, therefore, highly important to reinforce the Rainbow Resistance: Dandarah App feature that connects IPV survivors with a broad range of follow-up services already identified as SGM-welcoming and safe.

## 5. Strengths and Limitations

The current study included a convenience sample of SGM persons from six Brazilian states to assess the initial usability of the Rainbow Resistance: Dandarah App. Future work is needed to better understand the app’s usability among SGM prsons living in rural and hard-to-reach areas in Brazil. Future analysis is also required to evaluate if there are differences in the app’s usability based on demographic characteristics, sexual orientation, gender identity, and time since the most recent use of the panic button. It was not possible to examine those differences in the current study due to the convenience and sample size, which were not representative of the SGM community from Brazil.

It is also possible that SGM participants who already utilized the panic button may have different feedback than those who have never used this feature. Within the broad SGM community, transgender and gender non-conforming persons, as well as radicalized SGM persons, experience a higher prevalence of discrimination and violence, therefore, it is pivotal to consider unique barriers to receiving healthcare among these underserved populations as well. 

Violence and discrimination take many forms, and the lens of intersectionality is pivotal to allowing a better comprehension of the varied experiences of different members of the SGM community. The SGM community is made up of many distinct groups from across a broad spectrum of identities, but unfortunately, the lived realities, needs, and priorities of different groups within the SGM community are often not taken into consideration. Our pilot study was not developed to collect data that could identify which groups are experiencing higher levels of discrimination and violence among the SGM community from Brazil. However, based on our experience, we know that each individual within the broad Brazilian SGM community will experience different levels of vulnerability, discrimination, and violence based on other factors such as their ethnicity, socioeconomic status, disability, age, geographic location, gender identity, and sexual orientation, among others. For instance, Black, Indigenous, and other people of color who are members of the larger SGM community will face unique systemic challenges as a result of their intersecting identities. However, this exploratory study did not collect data tailored to discuss this aspect.

The SGM community experiences additional barriers to reporting experiences of discrimination and violence and accessing a broad range of services needed after an assault (e.g., post-rape care, shelters, psychological counseling, legal and social support, etc.). Therefore, innovative approaches to improve the reporting of discrimination/violence, such as the Rainbow Resistance: Dandarah App, are urgently needed for the SGM community.

Although we observed positive feedback on the usability of the Rainbow Resistance: Dandarah App, the results are preliminary, and further research is needed. The usability evaluation was conducted during the FGDs, with a moderator always present. The group setting and the presence of researchers may have inhibited participants’ responses. Therefore, it may be useful to assess the usability of the Rainbow Resistance: Dandarah App when there is no researcher present, through self-report as well. Additional evaluations should also include previously validated scales and a more robust design to identify the Dandarah App’s efficacy.

Despite these limitations, it is important to highlight that the intervention was conducted in one of the most dangerous countries for SGM persons to live in. Brazil records at least one murder of an SGM person every day, interventions are scarce, and reporting of violence/discrimination is almost absent due to police SGM phobia and victim blaming. To the best of our knowledge, this is the first nationwide intervention implemented in Brazil that allows SGM persons to report violence/discrimination and access a broad range of supportive services. Additionally, the Dandarah: Rainbow Resistance App includes crowdsourcing and mapping strategies, allowing users to identify services that are more welcoming and accessible for SGM persons. These additional features might be highly important to decrease barriers to accessing services among this highly vulnerable population.

## 6. Conclusions

The current study presented the development and initial usability evaluations of the Rainbow Resistance: Dandarah App; the findings suggest that the application is usable and fit for future effectiveness testing, filling a critical gap in treatment for SGM persons experiencing violence/discrimination in Brazil. Future work should assess the efficacy of the app, stratifying findings per demographic characteristics, sexual orientation, gender identity, and time since the most recent experience of discrimination/violence. It is also essential to evaluate the Dandarah App in connecting SGM persons with follow-up services, stratifying by type of discrimination/violence experienced and by subpopulation. The Rainbow Resistance: Dandarah App was recently incorporated within the Brazilian Ministry of Health core strategies to tackle the specific needs of the SGM population from Brazil, and we plan to continue to evaluate its efficacy over time.

## Figures and Tables

**Figure 1 ijerph-20-00280-f001:**
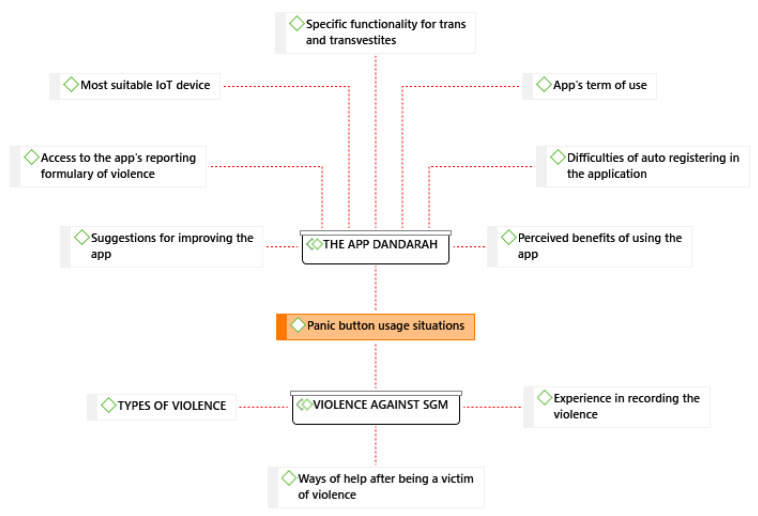
Thematic categories related to the Dandarah App features and violence toward SGM persons.

**Figure 2 ijerph-20-00280-f002:**
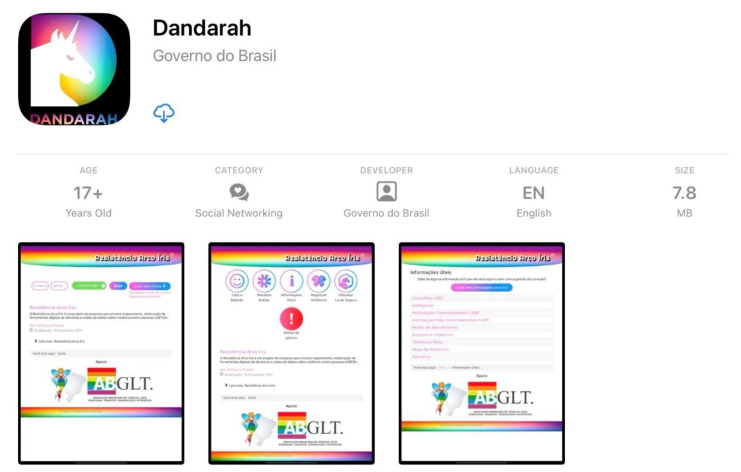
Rainbow Resistance: Dandarah App.

**Table 1 ijerph-20-00280-t001:** SGM suggestions to improve the Rainbow Resistance: Dandarah App.

Quotation	Dandarah Team Response	Themes	Density
*It gives you five number options [of emergency contacts after pressing the Panic Button], I think this panic button had to send you an alert to whoever has the app, so whoever is nearby helps you, regardless of who it is, because let’s suppose, that in my five numbers only there are people from City A, but I’m there in City B visiting someone. And then how am I going to call someone from City A to help me if I’m there in City B?*	SMS messages were included as an alternative, reducing the number of trusted contacts to three per user registered in the application.	Help after experiencing violence When to use the panic button	3
*I always talk about our health, not only physical but mental as well. Especially among the trans community. I’m worried about a suicide emergency, a suicide attempt, for example. That’s really common among trans people. So I think that maybe putting a hotline number on the ap or having some emergency numbers of psychologists and psychiatrists*	This strategy will be implemented in the following research study.	Help after experiencing violence When to use the panic button Types of violence	4
*We need to improve the app design. It needs to be more interactive, some buttons seem to be out of place… The settings button should go all the way down and the Panic Button should be the first one. It gotta be bigger and clear, that’s the most important thing*	The suggestion was implemented.	When to use the panic button Features important for trans and gender non-conforming App setting, terms, and conditions	4
*Like, it’s complicated… To access the Panic Button you have to scroll all the way down the page to appear it… Pan, pan, panic is all the way down, that’s wrong.*	The suggestion was implemented.	When to use the panic button	2
*Like, you only see the app symbol and the Panic Button…And what about everything else?*	Icons for other app sections were created, always with the panic button in highlight.	When to use the panic button	2
*So, about the bracelet [with a Panic Button], it will be cool to have at least two features. One red button to press right away, and another, if possible, to record verbal aggressions or something like that. Whoever is listening, will know what this person is telling you, and we would have proofs to fight for our rights, you know? You could have recorded pieces of evidence right there on your bracelet. That would be so awesome!*	Under development.	Internet of Things (IoT) devices are more adequate	2
*Like , those are the more important things: you gotta access the map and indicate [where the discrimination/violence is happening] and you don’t need lots of buttons, just one to complain and the Panic Button*	The Dandarah App has two sections: (1) recording safe places on a public map, in real time; (2) panic button more accessible on the initial screen.	When to use the panic button	2
*Like, we will record places safe and unsafe for us, right? Now I don’t know how the vulnerability issue will be handled. Who will have access to this information? I think that together with the Panic Button, you gotta add something else. Like a notification or stuff like that. It gotta become something else, you know?*	There is a map including all notifications, when the panic button was activated, and where. This information is managed by the research team.	When to use the panic button	2
*I think the Panic Button gotta be on the top. ‘Cas when you open the app, you will be in a panic situation*	The suggestion was implemented.	When to use the panic button	2
*I know that the Public Defender’s Office has night shifts on some days of the week… Maybe we could have an automatic notification whenever someone presses the Panic Button… Or it could be connected to a hotline or some other service that could follow up on this*	Not implemented yet.	When to use the panic button Violence reactions	3
*I think you gotta engage not only the population but also the police. Got it?*		When to use the panic button	2
*I think we could press the Panic Button when we witness someone being assaulted. You could have the option to record or take pictures*			2
*The Panic Button could have a timer. Like, you set up a timer, say I’ll meet someone for the first time… And I set up the timer for one hour. IF I don’t enter the app and turn it off, in one hour it automatically sends a message*			2
*An SMS could be sent from her App, like a voice file, so she could understand what’s going on…*			2
*I think if they could, like, make the panic button more accessible… Maybe every time you open the app you could have a question like: “* *Are you panicking, do you need help?”*			2
- *I have one more suggestion about the Panic Button… I don’t know if that’s possible* - *We could have a shortcut.* - *Yeap, a shortcut to your cell phone.*		Suggestions to improve the app	1
- *For example, if I pressed the Panic button several times, it was good to have the history, you know?* - *That’s right!* - *It was pressed such day, such time, then this other day and time… Like a history that I could have, just for me*	This history is sent to each user, as well as addresses and phones of supportive services.	When to use the panic button	2
*I think we could have access to numbers that we could call… Let’s say I was assaulted in city X, but I don’t know anyone there… If I press the Panic Button, it will send messages to my friends that live in City Y… But if I could look in the app and see, wow, I’m close to this place here, and that’s a place that my peers already pinned as queer-friendly, so I could search for the phone and look for help*	Addresses and phones of services from all over Brazil were included on the app, informed by the Brazilian Ministry of Health. These services were stratified by states to facilitate users’ searches.	Difficulties to register assaults in the app When to use the panic button	3
*After I press Panic Button, I gotta be able to see where I am, and what services are available around me.*		When to use the panic button	2

**Table 2 ijerph-20-00280-t002:** Sociodemographic characteristics of the Dandarah App users, Brazil, 2019–2022 *.

Sociodemographic Characteristics (*n* = 4114)	Frequency (%)
**Gender Identity**	
Cisgender men	2095 (50.9)
Transgender men	209 (5.1)
Cisgender women	788 (19.2)
Transgender women	238 (5.8)
*Travesti*	75 (1.8)
Other	477 (11.6)
NA	232 (5.6)
**Gender Identity “Other” (*n* = 477)**	
Non identified	163 (34.2)
Family/allies	20 (4.2)
Intersex	58 (12.2)
Non-binary	115 (24.1)
Genderqueer	4 (0.8)
NA	117 (24.5)
**Sexual Orientation**	
Bisexual	599 (14.6)
Gay	1788 (43.5)
Heterosexual	412 (10.0)
Homosexual	391 (9.5)
Lesbian	527 (12.8)
Pansexual	298 (7.2)
Other	99 (2.4)
**Sexual Orientation “Other” (*n* = 99)**	
Androsexual	2 (2.0)
Asexual	31 (31.3)
Non identified	14 (14.1)
Family/allies	1 (1.0)
Gynesexual	1 (1.0)
Other sexual expressions	11 (11.1)
Sexual orientation towards more than one gender	6 (6.1)
NA	33 (33.4)
**Intersex**	
Yes	280 (6.8)
**Race/Etnicity**	
Asian	73 (1.8)
White	1948 (47.3)
Indigenous	35 (0.9)
Mixed	1265 (30.7)
Black	670 (16.3)
Other	17 (0.4)
NA	106 (2.6)
**Age (years)**	
10–19	807 (19.6)
20–29	1699 (41.3)
30–39	861 (20.9)
40–49	434 (10.5)
50–59	105 (2.6)
≥60	20 (0.5)
NA	188 (4.6)
**Disability**	
Yes	93 (2.3)

* List of definitions included at the end of the manuscript.

## Data Availability

The data presented in this study are available on request from the corresponding author. The data are not publicly available due to privacy reasons.

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
