# Peer review of "The Dandarah App: An mHealth Platform to Tackle Violence and Discrimination of Sexual and Gender Minority Persons Living in Brazil"

_ijerph, 2022, doi:10.3390/ijerph20010280_

Round 1
Reviewer 1 Report
Thank you for addressing this very important topic. I found the manuscript interesting and look forward to reading future studies quantifying the efficacy of the APP.
The following references are on the reference list but not cited in the manuscript:
Ard & Makadon (2011)
Reuters (2022)
Rolle et al. (2018)
Author Response
REVIEWER 1
- Thank you for addressing this very important topic. I found the manuscript interesting and look forward to reading future studies quantifying the efficacy of the APP.
We thank the reviewer for their kind and constructive words.
2. The following references are on the reference list but not cited in the manuscript:
Ard & Makadon (2011)
Reuters (2022)
Rolle et al. (2018)
We revised the references and in-text citations accordingly. Thanks for your time and feedback.
Reviewer 2 Report
This manuscript describes the development of a mHealth solution to help address violence against Sexual and Gender Minority (SGM) in Brazil. This is a qualitative study where researchers obtained feedback from the SGM community from six states of Brazil after 24 months of app use. Twenty-two focus group discussions (FGDs) were conducted, and Thematic analysis was used to interpret the qualitative data. Participants found the intervention to be user-friendly and suggesting a few changes to the app. Findings suggest that the intervention is usable and fills an important gap in the wellbeing of SGM persons.
This is the first study to describe the effectiveness of mHealth solution to address violence against people and Sexual and Gender minorities from Brazil. This is a well-written manuscript and I applaud the authors for their hardwork.
Suggestions: There is a lack of uniformity in reference style.
Author Response
REVIEWER 2
1. This manuscript describes the development of a mHealth solution to help address violence against Sexual and Gender Minority (SGM) in Brazil. This is a qualitative study where researchers obtained feedback from the SGM community from six states of Brazil after 24 months of app use. Twenty-two focus group discussions (FGDs) were conducted, and Thematic analysis was used to interpret the qualitative data. Participants found the intervention to be user-friendly and suggesting a few changes to the app. Findings suggest that the intervention is usable and fills an important gap in the wellbeing of SGM persons.
This is the first study to describe the effectiveness of mHealth solution to address violence against people and Sexual and Gender minorities from Brazil. This is a well-written manuscript and I applaud the authors for their hard work.
We thank the reviewer for their positive comments.
2. Suggestions: There is a lack of uniformity in reference style.
We revised the reference list accordingly - thanks for the careful evaluation.
Reviewer 3 Report
Firstly, I would like to thank for the opportunity to read such interesting article which is focused on important topic.
The structure of the manuscript is relevant.
Abstract as well as Introduction section clearly describes the aims of the work and justified the need of such studies.
Both phases of applied methods of the study were also written relevantly. The description is clear and understable and enables similar studies to be repeated in the future.
In my subjective opinion, in the description of the methodology used, it is worth paying attention to the references to the concept of collective intelligence and netnography as a theoretical scientific framework for the CBPR methodology. This would allow for an increase in the number of scientific sources cited in the work. In my opinion, their current number - 21 - is sufficient, although more in-depth literature studies could be carried out.
The results section i salso written relevantly. The results are presented in a clear way. The reader is free to follow the authors' reasoning.
Limitations of the study as well as conclusions were also relevantly written.
In conclusion, I would like to recommend this manuscript for publication with the minor revisions suggested above.
Author Response
REVIEWER 3
Firstly, I would like to thank for the opportunity to read such interesting article which is focused on important topic.
The structure of the manuscript is relevant.
Abstract as well as Introduction section clearly describes the aims of the work and justified the need of such studies.
Both phases of applied methods of the study were also written relevantly. The description is clear and understandable and enables similar studies to be repeated in the future.
We appreciate the reviewer’s comments.
In my subjective opinion, in the description of the methodology used, it is worth paying attention to the references to the concept of collective intelligence and netnography as a theoretical scientific framework for the CBPR methodology. This would allow for an increase in the number of scientific sources cited in the work. In my opinion, their current number - 21 - is sufficient, although more in-depth literature studies could be carried out.
We thank the review for their important suggestion. The methods section was revised accordingly, including more references.
The results section is also written relevantly. The results are presented in a clear way. The reader is free to follow the authors' reasoning.
Limitations of the study as well as conclusions were also relevantly written.
In conclusion, I would like to recommend this manuscript for publication with the minor revisions suggested above.
We thank the reviewer for their time and constructive suggestions.
Reviewer 4 Report
See Attach file

Author Response
REVIEWER 4
The study focused on Sexual and/or Gender Minorities (SGMs); these are commonly discriminated against and violent. The study conducted 22 focus group discussions (FGDs) by looking at a total of 300 SGM personnel in six Brazilian states. And the development of mobile health solutions to help address violence against SGM (development of the Rainbow Resistance: Dandarah App.). The results support that the interventions carried out in this study are available, so the main contribution of this study remains an important guideline literature for SGM to provide improvement options. The reviewer considered the topics and directions discussed in this study to be very interesting, the literature was well explored, and the complete content of the article was a reference article.
We thank the reviewer for their positive comments.
However, it is recommended that modifications be made to meet the requirements of a scientific nature for the following issues:
The description of the development process of the Rainbow Resistance: Dandarah App is not clear enough, and it is recommended to draw a production flowchart to reinforce the discussion. It is impossible to determine whether the Dandarah App is a general or medical (psychological) level app; additional clarification is required.
We thank the reviewer for their important feedback. The revised manuscript incorporates a better description of Dandarah development, highlighting that it is a general mobile app and not a health app. A health app is categorized by the FDA as mobile software that diagnoses, tracks, or treats disease - our app does not offer those tools.
What is the architecture of community-based participatory research (CBPR) methodology? Further explanation may be required.
A better description of our CBPR methodology was included.
It is suggested that a control group must be required for the evaluation of this experiment.
Our manuscript is describing Dandarah App development, our intent with this manuscript is to describe the pilot/formative steps included in the app development. Users are now being followed up and further analysis about the intervention effectiveness and acceptability will be developed later on. It was not our intention to conduct an RCT, but rather to share with readers how we developed this strategy in close collaboration with the LGBTQ+ community from Brazil. We thank the reviewer for their suggestion, and will conduct a proper evaluation of the Dandarah App in the near future.
Since the study was the result of convenience and sample size, it is suggested that further robustness testing should be conducted. Such results are sufficient scientific evidence to support the conclusions.
We thank the reviewer for their suggestion. However this was an exploratory/pilot study that collected mostly qualitative data to inform the app development. Following qualitative methods standards, we recruited new participants until data saturation was reached - data saturation is the point at which no new information or themes is observed in the data. Robustness studies are not used in qualitative studies, but when we conduct a quantitative evaluation of the app acceptability and efficacy, we will keep in mind the important suggestion from the reviewer.
Only descriptive statistical content lack inferential statistics to prove support.
This was a pilot, exploratory and qualitative study. Therefore it was not developed to make predictions based on sample and data collected. When we conduct a large RCT we will definitely include a proper sampling strategy and adequate sample size that allow our team to conduct multivariate regression analysis, using the adequate statistical strategies to draw conclusions about the LGBTQ+ population from Brazil. We thank the reviewer for they careful evaluation of our work.
In table 2, are some groups considered for consolidation? For example, in Sexual Orientation, "Other" (n=99), Androswhile is 2, Family/allies are 1, and Gynesexua is 1. These groupings are not statistically explanatory.
We thank the reviewer for their careful evaluation of our manuscript. This was a qualitative, exploratory study and no statistical analysis beyond this descriptive SES table was intended to be included (nor our sampling strategy and sample size allows for any additional statistical analysis). Table 2 describes how each participant describes themselves, avoiding aggregating variables (e.g. gay and homosexual were not combined intentionally). This strategy was conducted to allow our team access to a broad and diverse idea of how the population describes themselves. A description of each category was also included. We thank the reviewer for their time and highly important suggestions.
Reviewer 5 Report
This paper described the development of a mHealth solution to help
address violence against SGMs - Rainbow Resistance: Dandarah App, with a synthesis of key results and feedback from the SGM community after 24 months of app use. Twenty-two focus group discussions (FGDs) were conducted with SGM persons living in six Brazilian states: Bahia, Federal District, São Paulo, Rio de Janeiro, Minas Gerais, Sergipe and Pará. A total of 300 SGM persons
participated in those FGD. Thematic analysis was used to interpret the qualitative data.
This was a very interesting topic and I enjoyed reading this manuscript. In general, I found the study quite sound. It is in the spirit of improving the manuscript that I provide the following questions and/or recommendations:
Question #1:
You conducted your study with SGM persons living in six Brazilian states: Bahia, Federal District, São Paulo, Rio de Janeiro, Minas Gerais, Sergipe and Pará. Is there a particular state where more violence occurs? In other words, is the violence against SGM persons evenly distributed among these six states? If one particular state has higher incidents of violence, please explain why this may be the case.
Question #2:
As I read your paper, I could not help but wonder how helpful or not helpful the police are in these six Brazilian states. Are police in these areas required to complete an annual training that checks and corrects their potential biases against SGM individuals?
Question #3:
What is the theoretical foundation of your study? A strong and applicable theoretical foundation would strengthen your work.
Question #4 (Methods):
Regarding the themes identified in this study, did you establish reliability between coders? If so, how many coders were used? What was the final reliability percent?
Question #5:
On Pages 14 and 15, you listed several terms, such as Cisgender Men, Transgender Men, Cisgender Women, Transgender Women, Transvestite, Non-Identified, Intersex, Non-Binary, Queer, Pansexual, Androsexual, Asexual, Gynosexual, Other Sexual Expressions, etc. You cannot assume that the reader knows the distinctions between these terms. You should provide definitions and examples for each of these terms. Also, what does “Other” and “NA” mean?
Question #6:
Are some SGM individuals more likely to experience violence? For example, an SGM person that is also Black.
Other Issues:
· You should define what you mean by violence and discrimination
· Many of your citations are a bit dated. Add 3-5 more citations from the years 2021 and 2022.
· You should alphabetize in text citations.
On Page 1, you wrote:
(Blondeel et al., 2018; Ayhan et al., 2020).
Change to:
(Ayhan et al., 2020; Blondeel et al., 2018).
On Page 2, you wrote:
(Malta et al., 2020; Gomes de Jesus et al., 2020; Torres et al., 2021).
Change to:
(Gomes de Jesus et al., 2020; Malta et al., 2020; Torres et al., 2021).
Author Response
REVIEWER 5
This paper described the development of a mHealth solution to help address violence against SGMs - Rainbow Resistance: Dandarah App, with a synthesis of key results and feedback from the SGM community after 24 months of app use. Twenty-two focus group discussions (FGDs) were conducted with SGM persons living in six Brazilian states: Bahia, Federal District, São Paulo, Rio de Janeiro, Minas Gerais, Sergipe and Pará. A total of 300 SGM persons participated in those FGD. Thematic analysis was used to interpret the qualitative data.
This was a very interesting topic and I enjoyed reading this manuscript. In general, I found the study quite sound.
We thank the reviewer for their time and careful reading.
It is in the spirit of improving the manuscript that I provide the following questions and/or recommendations:
Question #1: You conducted your study with SGM persons living in six Brazilian states: Bahia, Federal District, São Paulo, Rio de Janeiro, Minas Gerais, Sergipe and Pará. Is there a particular state where more violence occurs? In other words, is the violence against SGM persons evenly distributed among these six states? If one particular state has higher incidents of violence, please explain why this may be the case.
The selected states represent a mix of large, urban areas (São Paulo and Rio de Janeiro), mixed urban and rural (Minas Gerais and Bahia), mostly rural areas (Sergipe and Pará) and the Federal District. There is a high prevalence of underreporting of both physical and non-physical violence towards SGM persons in Brazil - especially in hard to reach, rural areas. Data of reported crimes show a higher prevalence of violence is in rural, hard to reach states such as Pará and Sergipe, and an expected higher absolute number of reported crimes in most populated and urban states such as São Paulo and Rio de Janeiro. We included one paragraph briefly discussing the different scenarios in selected states. Thanks for the suggestion.
Question #2: As I read your paper, I could not help but wonder how helpful or not helpful the police are in these six Brazilian states. Are police in these areas required to complete an annual training that checks and corrects their potential biases against SGM individuals?
We thank the reviewer for their constructive feedback. Unfortunately SGM awareness trainings are not offered or required by Brazilian police officers. Episodes of discrimination and even violence when a SGM individual goes to the police station to report a crime are common, increasing the already high levels of underreported crimes. Our app was developed to provide an alternative reporting strategy, where victims and witness can report a crime without going physically to the police station. We thank the reviewer for their careful reading and constructive feedback.
Question #3: What is the theoretical foundation of your study? A strong and applicable theoretical foundation would strengthen your work.
We appreciate the reviewer's suggestion. We included a specific section describing the theoretical framework used on this study. In brief, the development of Rainbow Resistance-Dandarah app was guided by Social Cognitive Theory (SCT), which examines the process by which personal factors, environmental factors, and human behavior influence each other.
Question #4 (Methods): Regarding the themes identified in this study, did you establish reliability between coders? If so, how many coders were used? What was the final reliability percent?
We thank the reviewer for this important question. We included an additional section describing our qualitative analysis, describing the coding strategy (conducted independently by two researchers) and final inter-coder reliability (Cohen kappa >0.80).
Question #5: On Pages 14 and 15, you listed several terms, such as Cisgender Men, Transgender Men, Cisgender Women, Transgender Women, Transvestite, Non-Identified, Intersex, Non-Binary, Queer, Pansexual, Androsexual, Asexual, Gynosexual, Other Sexual Expressions, etc. You cannot assume that the reader knows the distinctions between these terms. You should provide definitions and examples for each of these terms. Also, what does “Other” and “NA” mean?
We agree, and decided to include a list of definitions in the end of the manuscript. NA mean ‘Not Available/missing’. Each variable included the possibility to choose an open-ended answer, if participants did not identify themselves with the available categories - those open ended answers were combined in the best way possible and answers that could not be combined were included as “other” - this explanation was included in our revised manuscript.
Question #6: Are some SGM individuals more likely to experience violence? For example, an SGM person that is also Black.
Violence takes many forms, and the lens of intersectionality is pivotal to allows a better comprehension of the varied experiences of different members of the SGM community. The SGM community is made up of many distinct groups from across a broad spectrum of identities, but unfortunately the lived realities, needs and priorities of different groups within the SGM community are often not taken into account. Our pilot study was not able to collect data that could identify what groups are experiencing higher levels of discrimination and violence among the SGM community from Brazil. But based in our experience, we know that each individual within the broad Brazilian SGM community will experience different levels of vulnerability, discrimination and violence based on other factors such as their ethnicity, socioeconomic status, disability, age, geographic location, gender identity and sexual orientation, among others. We agree with the reviewer that Black, Indigenous, and other People of Color who are a member of the larger SGM community will face unique systemic challenges as a result of their intersecting identities. However this pilot, exploratory study did not collect data tailored to discuss this aspect. We included one paragraph briefly describing how intersectionality is key to better understand the experience of discrimination and violence among SGM persons from Brazil, and we will continue collecting this data to inform more adequate interventions.
Other Issues:
You should define what you mean by violence and discrimination
We included the World Health Organization definition of violence and discrimination in our background section. Thanks for your important suggestion. The definitions included are:
According to the WHO, violence is the "the intentional use of physical force or power, threatened or actual, against oneself, another person, or against a group or community, that either results in or has a high likelihood of resulting in injury, death, psychological harm, maldevelopment, or deprivation.”
https://www.who.int/groups/violence-prevention-alliance/approach
The principle of non-discrimination seeks ‘…to guarantee that human rights are exercised without discrimination of any kind based on race, colour, sex, language, religion, political, or other opinion, national or social origin, property, birth or other status such as disability, age, marital and family status, sexual orientation and gender identity, health status, place of residence, economic and social situation’.
https://www.who.int/news-room/fact-sheets/detail/human-rights-and-health
Many of your citations are a bit dated. Add 3-5 more citations from the years 2021 and 2022.
Following your recommendation we included 6 additional citations published between 2021-22
You should alphabetize in text citations.
On Page 1, you wrote: (Blondeel et al., 2018; Ayhan et al., 2020). Change to: (Ayhan et al., 2020; Blondeel et al., 2018).
On Page 2, you wrote: (Malta et al., 2020; Gomes de Jesus et al., 2020; Torres et al., 2021). Change to: (Gomes de Jesus et al., 2020; Malta et al., 2020; Torres et al., 2021).
We revised all in text citation. Thanks for the sharp eyes and important suggestions!
Round 2
Reviewer 4 Report
Format the article according to the specifications of the journal
Author Response
We thank the reviewer for their careful evaluation of our revised manuscript. Please see below how we addressed the remaining concerns.
1. English language and style are fine/minor spell check required
We thank the reviewer for their careful reading of our manuscript. We conducted a thorough spelling and grammar check of the manuscript.
2. Are the methods adequately described? Can be improved
We included the following additional sections on our methods description:
Approach: description of the method utilized on this exploratory, qualitative study - community based participatory research
Thorough description of phase 1 and phase 2: detailed description of what we did, when, where
Theoretical framework: Description of Social Cognitive Theory, the framework utilized to guide our study
Analysis: How we conducted the qualitative analysis, included how quotes were coded and inter-coder reliability
Thanks for your collaboration to improve our final manuscript.